# Amniocentesis in Twin Pregnancies: Risk Factors of Fetal Loss

**DOI:** 10.3390/jcm11071937

**Published:** 2022-03-31

**Authors:** Ewelina Litwinska, Magdalena Litwinska, Bartosz Czuba, Agnieszka Gach, Sebastian Kwiatkowski, Przemyslaw Kosinski, Piotr Kaczmarek, Miroslaw Wielgos

**Affiliations:** 11st Department of Obstetrics and Gynecology, Medical University of Warsaw, 03-242 Warsaw, Poland; litwinska.magdalena@gmail.com (M.L.); miroslaw.wielgos@gmail.com (M.W.); 2Women’s Health, Faculty of Health Sciences in Katowice, Medical University of Silesia, 40-055 Katowice, Poland; bartosz.czuba@sonomed.net; 3Department of Genetics, Polish Mother’s Memorial Hospital Research Institute, 93-338 Lodz, Poland; agagach@tlen.pl; 4Clinical Department of Obstetrics and Gynecology, Pomeranian Medical University, 71-455 Szczecin, Poland; kwiatkowskiseba@gmail.com; 5Department of Obstetrics, Perinatology and Gynecology, Medical University of Warsaw, 03-242 Warsaw, Poland; pkosinski.mail@gmail.com; 6Department of Gynecology, Fertility and Fetal Therapy, Polish Mother’s Memorial Hospital Research Institute, 93-338 Lodz, Poland; kaczmarekpiotr1@gmail.com

**Keywords:** amniocentesis, twin pregnancies, miscarriage, fetal loss

## Abstract

This study aims to determine if second trimester amniocentesis in twin pregnancies provides a significant independent contribution in the prediction of miscarriage or fetal loss at any stage of pregnancy. This was a retrospective cohort study of women with twin gestations booked for routine prenatal care in four fetal medicine units in Poland in the years 2010–2020. The study population included: (1) twin pregnancies that underwent amniocentesis at 16–20 weeks’ gestation; (2) twin pregnancies that did not require any further testing and were followed-up routinely. Univariable and multivariable regression analysis was used to define which maternal and pregnancy characteristics provided a significant independent contribution in the prediction of miscarriage and fetal loss at any stage of pregnancy. In the study period, 2645 twin pregnancies were eligible for analysis. There were 144 cases of miscarriage defined as fetal loss of one or both twins before 24 weeks and 40 cases of intrauterine death of one or both twins after 24 weeks. A total number of 162 twin pregnancies underwent amniocentesis at 16–20 weeks’ gestation. The rate of miscarriage before 24 weeks and the rate of fetal loss at any stage of pregnancy in the group that underwent amniocentesis was 10.49% and 13.58%, respectively, compared to 5.11% and 6.52% that did not undergo amniocentesis. Multivariable regression analysis showed that factors providing a significant independent contribution in the prediction of miscarriage and fetal loss at any stage of pregnancy were monochorionicity (MC), large intertwin discordance in crown-rump length (CRL), low Pregnancy Related Plasma Protein (PAPP-A) MoM and nuchal translucency (NT) above 95th centile. Amniocentesis in twin pregnancies does not provide a significant contribution in the prediction of miscarriage or fetal loss at any stage of pregnancy.

## 1. Introduction

Invasive testing using amniocentesis or chorionic villus sampling is recommended in twin pregnancies whenever definitive karyotype or chromosomal microarray results are needed. However, data concerning the procedure-related risk in both singleton and twin pregnancies are limited. There is only one randomized study concerning the risk of miscarriage after second trimester amniocentesis in singleton pregnancies which showed that the risk is 1% higher in the group that had amniocentesis compared to controls [1]. Recent studies in singleton pregnancies have shown that the amniocentesis-related risk is low and not significantly different from the risk in women who did not have any procedure [2]. The data concerning the amniocentesis-related risk in twin gestations varies significantly [3,4,5,6,7,8,9,10,11,12,13,14,15,16,17,18,19]. A recent meta-analysis has shown that the risk of fetal loss following amniocentesis in twins is lower than reported previously and did not differ from the background risk in twin pregnancy not undergoing invasive prenatal testing [20]. Data standardization is essential for accurate patient counseling.

The objective of this study is to define if second trimester amniocentesis in twin pregnancies provides a significant independent contribution in the prediction of miscarriage and fetal loss at any stage of pregnancy.

## 2. Materials and Methods

This was a retrospective multicenter cohort study of all twin pregnancies booked for routine pregnancy care before 14 weeks’ gestation in four fetal medicine units in Poland in the years 2010–2020. In these centers, all women were offered an appointment at 11–13 weeks’ gestation in order to determine gestational age according to the measurement of crown-rump length (CRL) of the bigger twin, assess fetal anatomy and offer first trimester combined screening test for aneuploidies. The combined test was performed by certified sonographers according to the Fetal Medicine Foundation algorithm and incorporated patient’s individual background risk, measurement of fetal nuchal translucency thickness (NT) and maternal serum biomarkers: pregnancy related plasma protein A (PAPP-A) and free beta-hCG. Screen positive result was defined as risk greater or equal to 1 in 300 for trisomy 21, 18 and 13 in one or both twins. In this study women were not offered cell-free DNA screening for aneuploidy. The indications for amniocentesis were a high-risk result from the first trimester combined test (*n* = 97), high NT in one or both twins (*n* = 38), large intertwin discordance in CRL (*n* = 13), maternal request (*n* = 5), history of aneuploidy in the previous pregnancy (*n* = 4), parental carriership of Duchenne muscular dystrophy (*n* = 1), hemophilia (*n* = 2) and paternal balanced translocation (*n* = 2).

At the time of the 11–13 weeks’ scan, maternal demographic characteristics were recorded, including maternal age, height, weight, method of conception, smoking status, parity and history of medical disorder.

Patients with a high-risk result were referred to fetal medicine specialist or geneticist for counseling about options for management and risks related to invasive testing. In patients who opted for amniocentesis, an ultrasonographic examination was performed prior to each intervention. Fetal anatomy, possible developmental disorders, the position of placenta(s), and the exact intrauterine positions of the fetuses were evaluated, and then the site of entry was determined. Most procedures were performed using two separate needles (139/162; 85%) but some operators preferred to use the single needle technique (23/162; 15%). In all cases, both amniotic sacs were sampled. Technique involving two separate needles was performed transabdominally without injecting a dye. After the insertion of the first 22-gauge spinal needle (Cook Medical, Bloomington, IN, USA), 15 mL of fluid is aspirated for cytogenetic evaluation and the needle is withdrawn. Then, the second sac is tapped at a different location. Single needle technique was first described by Jeanty et al. [21]. Under ultrasound guidance, after identifying the membrane dividing the two amniotic cavities, a 22-gauge spinal needle (Cook Medical, Bloomington, IN, USA) is inserted into the first sac close to the septum. After aspiration of 15 mL of fluid and careful labeling of the container, the needle is inserted through the septum into the second sac. An initial 1 mL of fluid is aspirated and discarded (to lower the risk of contamination from the first sac) and then 15 mL is collected from the second sac. No drugs were administered, except for anti-D IgG prophylaxis in Rh-negative patients.

For data analysis, we excluded all cases where one or both twins suffered from major structural or chromosomal disorders. Additionally, we excluded monochorionic (MC) pregnancies that developed severe TTTS before 20 weeks’ gestation and required fetoscopic laser coagulation of placental anastomoses. In addition, women who underwent elective termination of pregnancy because of fetal anomalies were excluded from the study, also those who previously had undergone chorionic villus sampling. Patients eligible for the study were divided into two groups: invasive group that underwent amniocentesis and control group that was followed-up routinely. Miscarriage was defined as fetal or pregnancy loss before 24 weeks’ gestation. Intrauterine death was defined as death of one or both twins after 24 weeks’ gestation.

Statistical comparison of the maternal and pregnancy characteristics in the groups was performed using Kruskal–Wallis test and post hoc analysis for continuous variables and chi^2^-square test and Fisher’s exact test for categorical variables. We conducted univariable and multivariable logistic regression analysis in order to determine which maternal and pregnancy characteristics provided a significant independent contribution in the prediction of miscarriage and fetal loss at any stage of pregnancy. A *p* value < 0.05 was considered statistically significant and post hoc Bonferroni correction was used to adjust for multiple comparisons where necessary.

## 3. Results

In the study period, 2888 twin pregnancies were booked for routine pregnancy care in four fetal medicine centers in Poland, but 243 pregnancies were excluded from further analysis due to following reasons: (1) a major structural or chromosomal abnormality in one or both twins was diagnosed (*n* = 148), (2) were lost to follow-up or had incomplete data (*n* = 85), (3) TTTS developed prior to 20 weeks requiring laser surgery (*n* = 11), (4) had previously undergone chorionic villous sampling (*n* = 12). A total of 2645 twin pregnancies were eligible for analysis. There were 2043 dichorionic and 602 monochorionic twin pregnancies. Amniocentesis was performed in 162 cases. Maternal and pregnancy characteristics in the groups that had amniocentesis (115 DC twins and 47 MC twins) or did not have amniocentesis (1928 DC twins and 555 MC twins) are presented in Table 1.

In the case of DCDA twins, in the group that underwent amniocentesis, there was a significantly higher maternal age, more woman conceived by IVF, there were more cases of large intertwin discordance, the nuchal translucency above 95th centile and PAPP-A MoM was significantly lower. In the case of MCDA twins, the differences between the two groups were not statistically significant.

There were 144 cases of miscarriage before 24 weeks and 40 cases of intrauterine death after 24 weeks. The comparison of maternal characteristics according to the occurrence of miscarriage and intrauterine death is presented in Table 2.

The reasons for fetal loss/pregnancy loss before 24 weeks were intrauterine fetal death of one twin (IUD) (*n* = 52), IUD of both twins (*n* = 71), preterm premature rupture of membranes (PPROM) (*n* = 13) and cervical insufficiency (*n* = 8).

In the group of women that miscarried compared to those who delivered two live born babies, there was a statistically higher maternal age and weight, more women conceived by assisted conception (IVF), and there was a higher prevalence of chronic hypertension and diabetes. Similarly, in pregnancies that ended up with intrauterine death of one or both twins compared to those that delivered two live born babies, maternal weight was higher, more women conceived by assisted conception and there was a higher prevalence of diabetes and autoimmune disorders.

Univariable and multivariable regression analysis was used to determine which maternal and pregnancy characteristics provided a significant independent contribution in the risk of miscarriage (Table 3) and risk of fetal loss at any stage of pregnancy (Table 4).

Univariable regression analysis showed that maternal factors providing a significant independent contribution in the risk of miscarriage (before 24 weeks’ gestation) were advanced maternal age (OR = 1.102; 95%CI: 1.066–1.140), increased maternal weight (OR = 1.027; 95%CI: 1.016–1.038), conception by IVF (OR = 2.527; 95%CI: 1.742–3.664) and history of medical disorder including chronic hypertension (OR = 5.272: 95%CI: 2.945–9.438) and diabetes mellitus (OR = 3.600; 95%CI: 2.014–6.435). The same analysis showed that pregnancy characteristics providing a significant independent contribution in the risk of miscarriage were monochorionicity (OR = 3.240: 95%CI: 2.303–4.559), high intertwin discordance in CRL (OR = 1.418; 95%CI: 1.352–1.487), NT above the 95th centile (OR = 7.956; 95%CI: 5.335–11.866) and low PAPP-A MoM (OR = 0.017: 95%CI: 0.007–0.04). Subsequently, we conducted a multivariable regression analysis and after adjusting for confounding factors, we found that factors increasing the risk of miscarriage were maternal height (OR = 1.037; 95%CI: 1.002–1.073), history of autoimmune disorder (OR = 17.981; 95%CI: 1.166–277.252), monochorionicity (OR = 2.568; 95%CI: 1.65–3.998), high intertwin discordance in CRL (OR = 1.365; 95%CI: 1.256–1.483), low PAPP-A MoM (OR = 0.051; 95%CI: 0.021–0.122) and nuchal translucency above the 95th centile (OR = 7.170; 95%CI: 4.215–12.197). While analyzing the risk of fetal loss at any stage of pregnancy, univariable regression analysis showed that maternal factors contributing to the risk of fetal loss were advanced maternal age (OR = 1.092; 95%CI: 1.060–1.125), increased maternal weight (OR = 1.028; 95%CI: 1.018–1.038), conception by IVF (OR = 2.779; 95%CI: 2.002–3.856) and history of medical disorder including chronic hypertension (OR = 4.293; 95%CI: 2.443–7.545) and diabetes mellitus (OR = 3.776; 95%CI: 2.252–6.330). Pregnancy characteristics associated with the risk of fetal loss at any stage were monochorionicity (OR = 2.928; 95%CI: 2.155–3.979), high intertwin discordance in CRL (OR = 1.454; 95%CI: 1.389–1.523), nuchal translucency above 95th centile (OR = 6.411; 95%CI: 4.397–9.347), low PAPP-A MoM (OR = 0.075; 95%CI: 0.040–0.138). After adjusting for confounding factors, multivariable regression analysis showed that factors contributing in the risk of fetal loss at any stage of pregnancy were conception by IVF (OR = 1.753; 95%CI: 1.111–2.764), monochorionicity (OR = 2.370; 95%CI:1.598–3.515), high intertwin discordance in CRL (OR = 1.370; 95%CI: 1.273–1.473), low PAPP-A MoM (OR = 0.171; 95%CI: 0.093–0.314), and nuchal translucency above the 95th centile (OR = 6.037; 95%CI: 3.684–9.894).

The risk of miscarriage before 24 weeks and the risk of fetal loss at any stage of pregnancy in the control group was to 5.11% and 6.52%, respectively, compared to 10.49% and 13.58% in the group that underwent amniocentesis. Although, the risk of miscarriage and fetal loss at any stage of pregnancy is twice as high in the group that underwent amniocentesis compared to the group that did not undergo amniocentesis, multivariable regression analysis showed that amniocentesis itself did not provide a significant independent contribution in the risk of miscarriage or fetal loss at any stage of pregnancy.

## 4. Discussion

Main findings of this study

The results of our study show that the incidence of miscarriage and fetal loss at any stage of pregnancy is twice as high in the group that underwent amniocentesis compared to the group that did not require invasive testing. Maternal and pregnancy characteristics that are strongly associated with the risk of miscarriage were defined in our study.

For the appropriate counseling of patients, it is essential to differentiate between various elements that define the risk of miscarriage [22]. In our study, we excluded all pregnancies with major structural or chromosomal abnormalities in one or both twins that have an a priori high risk of miscarriage or spontaneous intrauterine death. Additionally, we excluded monochorionic twins that developed severe TTTS and required fetoscopic laser surgery before 20 weeks’ gestation. Multivariable regression analysis was conducted in order to make necessary adjustments for cofounding factors and define factors that provide a significant independent contribution in the risk of miscarriage and fetal loss at any stage of pregnancy. Results of our study showed that pregnancy factors providing a significant independent contribution in the risk of miscarriage and fetal loss at any stage of pregnancy are: monochorionicity, low PAPP-A MoM, nuchal translucency above the 95th centile and large intertwin discordance. Amniocentesis did not increase the risk of miscarriage in none of the groups.

Strengths and limitations of the study

The main strength of the study is inclusion of a large, unselected population of twin pregnancies from four fetal medicine centers that specialize in multiple gestations. Univariable and multivariable regression analysis defined which maternal and pregnancy factors provide a significant independent contribution in the risk of miscarriage and fetal loss at any stage of pregnancy. The main limitation of the study is its non-randomized and retrospective design.

Comparison of the findings with previous studies in the literature

To our knowledge this is one of a few studies that included a large population of twin pregnancies and after exclusion of pregnancies with defects used univariable and multivariable regression analysis to evaluate if amniocentesis in twins increases the risk of miscarriage. Cahill et al. presented a single-center study on amniocentesis in unselected population of twin pregnancies and reported that the risk of pregnancy loss before 24 weeks’ gestation after mid-trimester amniocentesis in twin pregnancies is 1.8%; however, authors did not exclude pregnancies with major chromosomal or structural defects [3]. Sperling et al. presented results of a large population data set and reported that in woman with twin pregnancies who are screen positive for aneuploidy (greater than or equal to 1/200 for trisomy 21, greater than or equal to 1/100 for trisomy 18), amniocentesis does not increase the risk of miscarriage further (OR 1.32; 95% CI, 0.66–1.91) [4]. A number of studies investigated the amniocentesis-related pregnancy loss in twin pregnancies [5,6,7,8,9,10,11,12,13,14,15,16,17], however only six studies included a control group [5,9,10,13,14,17]. Most of these studies found no significant difference in the rate of fetal loss between the amniocentesis group and controls [5,9,10,13]. Toth-Pal et al. reported that genetic amniocentesis increases the risk of miscarriage but only within the first four weeks following the procedure [14]. Yukobowich et al. reported a statistically significant differences with a fetal loss rate of 2.7% in the amniocentesis group versus 0.6% in the control group [17]. However, due to small number of patients in these studies and heterogenous inclusion criteria, it is difficult to draw conclusions. There are also several meta-analyses that aimed to estimate the risk of amniocentesis in twins [18,19,20]. Agarwal et al. conducted a systematic review on pregnancy loss in twins after genetic amniocentesis and found that the overall pregnancy loss in twins was 3.07% (95% CI, 1.83–4.61) vs. 1.9% (95% CI, 1.4–2.5) in singletons. The authors concluded that patients should be counseled that the risk of miscarriage increases by approximately 1%, over and above the background risk, after amniocentesis in twins [19]. Our study shows that this excess risk is not entirely due to the invasive procedure but to some extent the demographic and pregnancy characteristics of the patient undergoing amniocentesis. Our results are similar to those published recently by Elger T et al., who aimed to estimate the chorionic villous sampling (CVS) related risk of fetal loss in twin pregnancies after adjustment for cofounding factors. The authors reported that in twin pregnancies undergoing CVS, there is a trend for an increased risk of fetal loss from CVS after adjustment for maternal and pregnancy characteristics but this does not reach statistical significance [23].

## 5. Conclusions

Although the rate of miscarriage and fetal loss at any stage of pregnancy in the group that underwent amniocentesis is twice as high compared to the group that did not undergo amniocentesis, amniocentesis in twin pregnancies does not provide a significant contribution in the prediction of miscarriage or fetal loss at any stage of pregnancy.

## Figures and Tables

**Table 1 jcm-11-01937-t001:** Maternal and pregnancy characteristics in the study groups.

Characteristic	DCDA		MCDA	
	No Amniocentesis (*n* = 1928)	Amniocentesis (*n* = 115)	No Amniocentesis (*n* = 555)	Amniocentesis (*n* = 47)
Age (years)	31.6 (28.2–35.1)	35.7 (32.0–38.2) *	32 (28.1–35.8)	32 (29.8–36.6) ^NS^
Weight (kg)	68.3 (61.5–75.5)	67 (60.8–76) ^NS^	68 (61.3–77)	65 (56–79) ^NS^
Height (cm)	167 (163–172)	165.1 (160–172.7) ^NS^	166 (162–171)	165 (160–172) ^NS^
Conception				
Spontaneous	1641 (86.1%)	88 (76.5%) ^NS^	448 (80.7%)	43 (91.5%) ^NS^
IVF	287 (14.9%)	27 (23.5%)	107 (19.3%)	4 (8.5%) ^NS^
Cigarette smoker	24 (1.2%)	0 (0%) ^NS^	6 (1.1%)	0 (0%) ^NS^
History of medical disorder				
Chronic hypertension	53 (2.7%)	0 (0%) ^NS^	21 (3.8%)	0 (0%) ^NS^
Diabetes mellitus	59 (3.1%)	0 (0%) ^NS^	38 (6.8%)	0 (0%) ^NS^
SLE/APS	3 (0.2%)	0 (0%) ^NS^	0 (0%)	0 (0%) ^NS^
Parity				
Nulliparous	1001 (51.9%)	56 (48.7%) ^NS^	274 (49.4%)	29 (61.7%) ^NS^
Multiparous	927 (48.1%)	59 (51.3%) ^NS^	281 (50.6%)	18 (38.3%) ^NS^
Intertwin discordance in CRL				
≥10%	35 (1.8%)	19 (16.5%) *	32 (5.8%)	6 (12.8%) ^NS^
≥20%	0 (0%)	1 (0.9%) ^†^	5 (0,9%)	0 (0%) ^NS^
Nuchal translucency				
>95th percentile	104 (5.4%)	12 (10.4%) ^†^	50 (9.0%)	6 (12.8%) ^NS^
PAPP-A MoM	1.09 (0.74–1.55)	0.94 (0.75–1.25) ^†^	1.19 (0.69–1.54)	0.87 (0.7–1.36) ^NS^

MCDA: monochorionic twins; DCDA: dichorionic twins; IVF: in vitro fertilization; SLE: systemic lupus erythematosus; APS: antiphospholipid syndrome; CRL: crown rump length. Values given as *n* (%) or median (interquartile range). Significance level: * *p* < 0.0001; ^†^ *p* < 0.01; ^NS^—not significant.

**Table 2 jcm-11-01937-t002:** Comparison of maternal characteristics according to the occurrence of miscarriage and IUD.

Characteristic	Live Birth (*n* = 2461)	Miscarriage (*n* = 144)	IUD (*n* = 40)
Age (years)	31.7 (28.2–35.3)	34.8 (31.4–38.3) *	32.8 (29.7–36,2) ^NS^
Weight (kg)	68 (61.2–75.4)	73.3 (65.5–84.9) *	72.5 (66.15–84.9) ^†^
Height (cm)	166 (162–171)	168 (163–172) ^NS^	167 (162.5–171) ^NS^
Conception			
Spontaneous	2096 (85.2%)	100 (69.4%)	16 (40%)
IVF	365 (14.8%)	44 (30.6%) *	24 (60%)*
Cigarette smoker	27 (1.1%)	3 (2.1%) ^NS^	0 (0%) ^NS^
History of medical disorder			
Chronic hypertension	57 (2.3%)	16 (11.1%) *	1 (2,5%) ^NS^
Diabetes mellitus	77 (3.1%)	14 (10.4%) *	5 (12.5%) *
SLE/APS	2 (0.1%)	1 (0.7%) ^NS^	0 (0%) ^†^
Parity			
Nulliparous	1274 (51.8%)	69 (47.9%) ^NS^	17 (42.5%) ^NS^
Multiparous	1187 (48.2%)	75 (52.1%) ^NS^	23 (57.5%) ^NS^

IUD: intrauterine death; SLE: systemic lupus erythematosus; APS: antiphospholipid syndrome. Significance level: * *p* < 0.0001; ^†^
*p* < 0.01; ^NS^—not significant; Significance level after Bonferroni correction *p* = 0.025; Values given as *n* (%) or median (interquartile range).

**Table 3 jcm-11-01937-t003:** Univariable and multivariable regression analysis to assess contribution from maternal and pregnancy characteristics and independent contribution of amniocentesis in prediction of miscarriage.

Characteristic	Univariate		Multivariate	
	OR (95% CI)	*p*	OR (95% CI)	*p*
Age (years)	1.102 (1.066; 1.140)	0.000	1.036 (0.988; 1.087)	0.145
Weight (kg)	1.027 (1.016; 1.038)	0.000	1.006 (0.990; 1.022)	0.459
Height (cm)	1.022 (0.995; 1.049)	0.105	1.037 (1.002; 1.073)	0.040
Conception				
Spontaneous (reference)	1.0	-	1.0	-
IVF	2.527 (1.742; 3.664)	0.000	1.502 (0.885; 2.550)	0.132
Cigarette smoker	1.918 (0.575; 6.399)	0.289	0.571 (0.070; 4.682)	0.602
History of medical disorder				
Chronic hypertension	5.272 (2.945; 9.438)	0.000	2.081 (0.892; 4.851)	0.090
Diabetes mellitus	3.600 (2.014; 6.435)	0.000	1.529 (0.676; 3.458)	0.308
SLE/APS	8.545 (0.77; 94.804)	0.081	17.981 (1.166; 277.252)	0.038
Parity				
Nulliparous	0.737 (0.288; 1.886)	0.524	1.936 (0.500; 7.497)	0.339
Intertwin discordance in CRL	1.418 (1.352; 1.487)	0.000	1.365 (1.256; 1.483)	0.000
≥10%	35.630 (21.618; 58.726)	0.000	1.137 (0.444; 2.916)	0.789
≥20%	88.489 (10.268; 762.606)	0.000	0.228 (0.019; 2.776)	0.246
Nuchal translucency				
>95th percentile	7.956 (5.335; 11.866)	0.000	7.170 (4.215; 12.197)	0.000
PAPP-A MoM	0.017 (0.007; 0.041)	0.000	0.051 (0.021; 0.122)	0.000
Amniocentesis	1.311 (0.954; 1.802)	0.094	1.705 (0.630; 4.612)	0.294
Chorion				
DCDA (reference)	1.0	-	1.0	-
MCDA	3.240 (2.303; 4.559)	0.000	2.568 (1.65; 3.998)	0.000

IVF: in vitro fertilization; SLE: systemic lupus erythematosus; APS: antiphospholipid syndrome; CRL: crown rump length; MCDA: monochorionic twins; DCDA: dichorionic twins.

**Table 4 jcm-11-01937-t004:** Univariate and multivariate regression analysis to assess contribution from maternal and pregnancy characteristics and independent contribution of amniocentesis in prediction of fetal loss at any stage.

Characteristic	Univariate		Multivariate	
	OR (95% CI)	*p*	OR (95% CI)	*p*
Age (years)	1.092 (1.060; 1.125)	0.000	1.026 (0.983; 1.070)	0.235
Weight (kg)	1.028 (1.018; 1.038)	0.000	1.011 (0.996; 1.025)	0.148
Height (cm)	1.018 (0.994; 1.042)	0.140	1.024 (0.993; 1.055)	0.129
Conception				
Spontaneous (reference)	1.0	-	1.0	-
IVF	2.779 (2.002; 3.856)	0.000	1.753 (1.111; 2.764)	0.016
Cigarette smoker	1.494 (0.449; 4.972)	0.513	0.543 (0.076; 3.891)	0.544
History of medical disorder				
Chronic hypertension	4.293 (2.443; 7.545)	0.000	1.726 (0.771; 3.862)	0.184
Diabetes mellitus	3.776 (2.252; 6.330)	0.000	1.485 (0.713; 3.093)	0.291
SLE/APS	6.678 (0.603; 73.992)	0.122	13.902 (0.911; 212.148)	0.058
Parity				
Nulliparous	0.510 (0.246; 1.057)	0.070	1.122 (0.341; 3.694)	0.85
Intertwin discordance in CRL	1.454 (1.389; 1.523)	0.000	1.370 (1.273; 1.473)	0.000
≥10%	38.875 (24.328; 62.121)	0.000	1.708 (0.756; 3.857)	0.198
≥20%	68.715 (7.985; 591.343)	0.000	0.108 (0.009; 1.250)	0.075
Nuchal translucency				
>95th percentile	6.411 (4.397; 9.347)	0.000	6.037 (3.684; 9.894)	0.000
PAPP-A MoM	0.075 (0.040; 0.138)	0.000	0.171 (0.093; 0.314)	0.000
Amniocentesis	2.368 (0.914; 6.139)	0.076	1.274 (0.498; 3.258)	0.614
Chorion				
DCDA (reference)	1.0	-	1.0	-
MCDA	2.928 (2.155; 3.979)	0.000	2.370 (1.598; 3.515)	0.000

IVF: in vitro fertilization; SLE: systemic lupus erythematosus; APS: antiphospholipid syndrome; CRL: crown rump length; MCDA: monochorionic twins; DCDA: dichorionic twins.

## Data Availability

The datasets used and analyzed during the current study are available from the corresponding author on reasonable request.

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
