# Peer review of "Amniocentesis in Twin Pregnancies: Risk Factors of Fetal Loss"

_jcm, 2022, doi:10.3390/jcm11071937_

Round 1
Reviewer 1 Report
In this retrospective cohort study, authors analyzed if second trimester amniocentesis in twin pregnancies provides a significant independent contribution to miscarriage/fetal loss prediction during pregnancy. Even though novelty of the study can be considered as average, overall merit is high. This reviewer has no concerns about the quality of the manuscript.
In this study, Litwinska et al, reported their data on the retrospective cohort study on the twin pregnancies and authors questioned if second trimester amniocentesis provides a significant independent contribution to the prediction of miscarriage and fetal loss in mentioned group. Depending on the collected and analyzed data, authors addressed that several maternal and pregnancy characteristics are strongly associated with the prediction of pregnancy complications where amniocentesis is not reported as one of them. Even though the topics originality is questionable, the design of the analysis, the inclusion/exclusion criteria, overall results of this study provide well organized, strong data about the possible risk factors that may contribute to miscarriage and fetal loss in twin pregnancies. This is a well written, reader friendly manuscript that addresses the main question. However, I can suggest authors to re-write the title of the manuscript not to cause a misunderstanding to what they report in this manuscript.Author Response
Reviewer 1 report:
In this retrospective cohort study, authors analysed if second trimester amniocentesis in twin pregnancies provides a significant independent contribution to miscarriage/fetal loss prediction during pregnancy. Even though novelty of the study can be considered as average, overall merit is high. This reviewer has no concerns about the quality of the manuscript.
Response: Thank you.
In this study, Litwinska et al, reported their data on the retrospective cohort study on the twin pregnancies and authors questioned if second trimester amniocentesis provides a significant independent contribution to the prediction of miscarriage and fetal loss in mentioned group. Depending on the collected and analyzed data, authors addressed that several maternal and pregnancy characteristics are strongly associated with the prediction of pregnancy complications where amniocentesis is not reported as one of them. Even though the topics originality is questionable, the design of the analysis, the inclusion/exclusion criteria, overall results of this study provide well organized, strong data about the possible risk factors that may contribute to miscarriage and fetal loss in twin pregnancies. This is a well written, reader friendly manuscript that addresses the main question. However, I can suggest authors to re-write the title of the manuscript not to cause a misunderstanding to what they report in this manuscript.
Response: Thank you. The title of the manuscript has now been changed.
Reviewer 2 Report
The present paper is well written and addresses an actual topic in fetal medicine.
I would strongly suggest adding a comment on a very similar paper from Elger T et al. (Fetal loss after chorionic villus sampling in twin pregnancy. Ultrasound Obstet Gynecol. 2021 Jul;58(1):48-55. doi: 10.1002/uog.23694. PMID: 34038977) which is extremely pertinent to the present work and cannot be uncited.
METHODS: please specify why the Authors decided to use a non parametric test for multiple groups comparison (Kruskal-Wallis)
RESULTS:
Table 1- It would be interesting to see the statistical differences between y/n amniocentisis groups by means of p-value or by underlying the significant differences
Page 4. I would put the sentence ‘There were 144 cases of miscarriage before 24 weeks and 40 cases of intrauterine death after 24 weeks’ before the sentence ‘The comparison of maternal characteristics according to the occurrence of miscarriage and intrauterine death is presented in table 2.’ and table 2.
Table 3. The fact that maternal height becomes significantly associated to the main outcome at the multivariable analysis makes the results a bit questionable since there is no significance at the univariate analysis nor a concrete theoretical explanation. How can the Authors explain this unexpected result and the reliability of their analysis?
Author Response
Reviewer 2 report:
The present paper is well written and addresses an actual topic in fetal medicine.
Response: Thank you.
I would strongly suggest adding a comment on a very similar paper from Elger T et al. (Fetal loss after chorionic villus sampling in twin pregnancy. Ultrasound Obstet Gynecol. 2021 Jul;58(1):48-55. doi: 10.1002/uog.23694. PMID: 34038977) which is extremely pertinent to the present work and cannot be uncited.
Response: Thank you. A comment on the paper by Elger T et al. has been added in the section Discussion (Comparison of the findings with previous studies in the literature).
METHODS: please specify why the Authors decided to use a non parametric test for multiple groups comparison (Kruskal-Wallis)
Response: Thank you. We decided to choose this test because the variables did not have a normal distrubution. This fact was tested using the Shapiro-Wilk test.
RESULTS:
Table 1- It would be interesting to see the statistical differences between y/n amniocentisis groups by means of p-value or by underlying the significant differences
Response: Thank you. This information has now been added in Table 1 and a comment is added in the section Results.
Page 4. I would put the sentence ‘There were 144 cases of miscarriage before 24 weeks and 40 cases of intrauterine death after 24 weeks’ before the sentence ‘The comparison of maternal characteristics according to the occurrence of miscarriage and intrauterine death is presented in table 2.’ and table 2.
Response: Thank you. The order of the sentences has now been changed accordingly.
Table 3. The fact that maternal height becomes significantly associated to the main outcome at the multivariable analysis makes the results a bit questionable since there is no significance at the univariate analysis nor a concrete theoretical explanation. How can the Authors explain this unexpected result and the reliability of their analysis?
Response: Thank you. We agree that this finding is questionable. However, we decided to incorporate all factors in the multivarible logistic regression and not only those that were statistically important in the univaribale logistic regression analysis. When we compared our results with previously published papers we found that some other authors also found that maternal height was significantly associated with the main outcome (example Beta et al.).
Reviewer 3 Report
This manuscript presents amniocentesis-related risk in twin pregnancies. This study performed by retrospective multicenter-cohort and finally, included 2,645 twin pregnancies at four medicine units in Poland between 2010 ~ 2020.
The authors report the rate of miscarriage before 24 weeks and the rate of fetal loss at any stage of pregnancy in the group that underwent amniocentesis was 10.49% and 13.58% respectively, compared to 5.11% and 6.52% that did not undergo amniocentesis. Multivariable regression analysis showed that factors providing a significant independent contribution in the prediction of miscarriage and fetal loss at any stage of pregnancy were MC twin, large intertwin discordance in CRL, low PAPP-A MoM and fetal NT above 95th centile.
Therefore, the authors assert that amniocentesis in twin pregnancies does not provide a significant contribution in the prediction of miscarriage or fetal loss at any stage of pregnancy.
Although the need for amniocentesis has decreased a lot recently due to NIPT in twin pregnancies, when diagnostic test is required at the beyond of the gestational week to be performed by CVS, amniocentesis is still being widely performed. From this point of view, this is interesting and helpful research.
However, I have considerable doubts about the results of this article.
1. Too high rate of miscarriage before 24 weeks in the group underwent amniocentesis : 10.49%
Although authors explained that this excess risk is not entirely due to the invasive procedure but to some extent the demographic and pregnancy characteristics of the patient undergoing amniocentesis, In my opinion, even though chromosome and structural abnormalities were excluded, if this high risk is true, it can be considered unreliable in drawing conclusions from it.
2. The authors should describe the indications of amniocentesis in more detail. This is because it is thought that the fetal loss rate after the procedure may differ depending on the indication of the amniocentesis.
3. The authors should describe the situation of the miscarried cases in more detail. Is it due to PPROM, IIOC, FDIU, or only one fetus is miscarried?
Author Response
Reviewer 3 report:
This manuscript presents amniocentesis-related risk in twin pregnancies. This study performed by retrospective multicenter-cohort and finally, included 2,645 twin pregnancies at four medicine units in Poland between 2010 ~ 2020.
The authors report the rate of miscarriage before 24 weeks and the rate of fetal loss at any stage of pregnancy in the group that underwent amniocentesis was 10.49% and 13.58% respectively, compared to 5.11% and 6.52% that did not undergo amniocentesis. Multivariable regression analysis showed that factors providing a significant independent contribution in the prediction of miscarriage and fetal loss at any stage of pregnancy were MC twin, large intertwin discordance in CRL, low PAPP-A MoM and fetal NT above 95th centile.
Therefore, the authors assert that amniocentesis in twin pregnancies does not provide a significant contribution in the prediction of miscarriage or fetal loss at any stage of pregnancy.
Although the need for amniocentesis has decreased a lot recently due to NIPT in twin pregnancies, when diagnostic test is required at the beyond of the gestational week to be performed by CVS, amniocentesis is still being widely performed. From this point of view, this is interesting and helpful research.
Response: Thank you
However, I have considerable doubts about the results of this article.
- Too high rate of miscarriage before 24 weeks in the group underwent amniocentesis : 10.49%
Although authors explained that this excess risk is not entirely due to the invasive procedure but to some extent the demographic and pregnancy characteristics of the patient undergoing amniocentesis, In my opinion, even though chromosome and structural abnormalities were excluded, if this high risk is true, it can be considered unreliable in drawing conclusions from it.
Response: Thank you. Initially we were also surprised about the relatively high rate of fetal loss before 24 weeks in the group that underwent amniocentesis. A recent study on 6225 twin pregnancies showed that overall the risk of miscarriage in an unselected population is between 2.3% (DCDA twins) and 7.7%(MCDA twins). However, in our study the prevalence of important risk factors of miscarriage was significantly higher in the amniocentesis group compared to the group that was follow-up routinely. It is true that case-control studies are showing a much lower rate of miscarriage in the amniocentesis group however a small number of patients in these studies precludes any meaningful conclusions to be drawn. If we look at more valuable papers that include unselected cohort of twins that either received invasive testing or were followed-up routinely, a different numbers emerge: in the study by Sperling et al. the loss rate in woman who underwent amniocentesis was 8.8%; similarly in a recent study by Elger et al. the rate of fetal loss after first trimester invasive testing was 8.5%. Additionally, I can assure you about the reliability of our data since they were collected and archived using a unified software in all included centres.
- The authors should describe the indications of amniocentesis in more detail. This is because it is thought that the fetal loss rate after the procedure may differ depending on the indication of the amniocentesis.
Response: Thank you. A detailed description of the indications for amniocentesis is now added in section Materials and methods.
- The authors should describe the situation of the miscarried cases in more detail. Is it due to PPROM, IIOC, FDIU, or only one fetus is miscarried?
Response: Thank you. A detailed description of the situation of the miscarried cases is now added in section Results. However, in some of these cases the leading reason for miscarriage was not certain since the patient presented in emergency department with signs of incomplete abortion.
Reviewer 4 Report
This manuscript presents interesting data on pregnancy losses in twin pregnancies in Poland in a last decade. In this study authors show contribution of independent risk factors with a large population of twin pregnancies using univariate and multivariate regression analysis. Although the data showed a doubling of the pregnancy loss rate for twin pregnancies, the authors concluded that fetal loss in twin pregnancies was not significantly increased by amniocentesis.
Author Response
Reviewer 4 report:
This manuscript presents interesting data on pregnancy losses in twin pregnancies in Poland in a last decade. In this study authors show contribution of independent risk factors with a large population of twin pregnancies using univariate and multivariate regression analysis. Although the data showed a doubling of the pregnancy loss rate for twin pregnancies, the authors concluded that fetal loss in twin pregnancies was not significantly increased by amniocentesis.
Response: Thank you
Round 2
Reviewer 3 Report
Thank you for the authors' sincere answers.
Despite the authors' answers, the fact that the miscarriage rate after amniocentesis is too high compared to other reports remains unchanged that it doubts the reliability of the research results, and it is worrisome that it could give readers information that one in ten women with twin pregnancy could be miscarried if they underwent amniocentesis.
Agarwal K, Alfirevic Z. Pregnancy loss after chorionic villus sampling and genetic amniocentesis in twin pregnancies: a systematic review. Ultrasound Obstet Gynecol 2012;40:128–34.
Simonazzi G, Curti A, Farina A, Pilu G, Bovicelli L, Rizzo N. Amniocentesis and chorionic villus sampling in twin gestations: which is the best sampling technique?. Am J Obstet Gynecol 2010;202:365.e1–5.
Cahill AG, Macones GA, Stamilio DM, Dicke JM, Crane JP, Odibo AO. Pregnancy loss rate after mid-trimester amniocentesis in twin pregnancies. Am J Obstet Gynecol 2009;200:257.e1–6.
Furthermore, the recently reported ACOG Practice Bulletin, the procedure-associated pregnancy loss rates for both amniocentesis and CVS in twins are similar (reported at 1~1.8%) and are slightly increased compared with loss rates reported in women with singleton gestations (Obstetrics and Gynecology 2021 VOL. 137, NO. 6, e145).
For this reason, I'd like to thank the authors for their hard work. It is recommended to analyze the data again to obtain true procedure-associated petal loss rates.
Author Response
Thank you for the suggestions. We very much appreciate reviewers effort to improve our paper. We were asked to reanalyse the data in order to achieve results that are comparable with previously published literarture. However, the study design of these papers differs from ours. Our study is one of very few studies that presented a large cohort of unselected twin pregancies and analysed factors that provide an independent contribution in the risk of miscarriage. In this study we did not aim to give a procedure-related risk of miscarriage which is cited in the papers pointed by the reviewer as examples of data that are inconsistent with our study. Our aim was to evaluate factors that provide a significant independent contribution in the risk of miscarriage. This is a different way of evaluating the risks associated with twin pregnancy and allows to give a new insight in this subject. Our study could be compared with a recently published paper by the Nicolaides group. In this study that included a population of 8581 twin pregnancies the rate of miscarriage before 24 weeks in the CVS group was 8.5% [Elger T, Akolekar R, Syngelaki A, De Paco Matallana C, Molina FS, Gallardo Arozena M, Chaveeva P, Persico N, Accurti V, Kagan KO, Prodan N, Cruz J, Nicolaides KH. Ultrasound Obstet Gynecol. 2021, 58, 48-55]. Moreover, we are sorry to hear that the reader could get an impression that amniocentesis is twin pregnancies is related to high risk of miscarriage. Our results, which are based on reliable statistical analysis, proved that amniocentesis itself does not increase the risk of miscarriage. We clearly defined factors that are associated with high risk of miscarriage and amniocentesis is not one of them. We hope that the reviewer accepts our arguments and supports publication of our data.